# Circadian Synchronization of Feeding Attenuates Rats’ Food Restriction-Induced Anxiety and Amygdalar Thyrotropin-Releasing Hormone Downregulation

**DOI:** 10.3390/ijms25115857

**Published:** 2024-05-28

**Authors:** Paulina Soberanes-Chávez, Jariz Trujillo-Barrera, Patricia de Gortari

**Affiliations:** 1Laboratorio de Neurofisiología Molecular, Dirección de Neurociencias, Instituto Nacional de Psiquiatría Ramón de la Fuente Muñiz, Ciudad de México 14370, Mexico; paulinasoberanes@inprf.edu.mx (P.S.-C.); jariz.trujillo@ednissste.com.mx (J.T.-B.); 2Escuela de Dietética y Nutrición del ISSSTE, Ciudad de México 14070, Mexico

**Keywords:** circadian feeding, anxiety, TRH, intermittent fasting, amygdala, corticosterone, stress

## Abstract

Anxiety is a common comorbidity of obesity, resulting from prescribing long-term caloric restriction diets (CRDs); patients with a reduced food intake lose weight but present anxious behaviors, poor treatment adherence, and weight regain in the subsequent 5 years. Intermittent fasting (IF) restricts feeding time to 8 h during the activity phase, reducing patients’ weight even with no caloric restriction; it is unknown whether an IF regime with ad libitum feeding avoids stress and anxiety development. We compared the corticosterone blood concentration between male Wistar rats fed ad libitum or calorie-restricted with all-day or IF food access after 4 weeks, along with their anxiety parameters when performing the elevated plus maze (EPM). As the amygdalar thyrotropin-releasing hormone (TRH) is believed to have anxiolytic properties, we evaluated its expression changes in association with anxiety levels. The groups formed were the following: a control which was offered food ad libitum (C-adlib) or 30% of C-adlib’s energy requirements (C-CRD) all day, and IF groups provided food ad libitum (IF-adlib) or 30% of C-adlib’s requirements (IF-CRD) with access from 9:00 to 17:00 h. On day 28, the rats performed the EPM and, after 30 min, were decapitated to analyze their amygdalar TRH mRNA expression by in situ hybridization and corticosterone serum levels. Interestingly, circadian feeding synchronization reduced the body weight, food intake, and animal anxiety levels in both IF groups, with ad libitum (IF-adlib) or restricted (IF-CRD) food access. The anxiety levels of the experimental groups resulted to be negatively associated with TRH expression, which supported its anxiolytic role. Therefore, the low anxiety levels induced by synchronizing feeding with the activity phase would help patients who are dieting to improve their diet therapy adherence.

## 1. Introduction

Obesity prevalence is alarmingly increasing worldwide in part because patients develop neuropsychiatric comorbidities, including anxiety and depression [1,2]. Anxiety may be developed in patients with obesity due to different factors, such as disruptions of eating habits [3,4] with consequent alterations in their sleep/awake patterns [5,6], high-fat diet-induced neuroinflammation [7,8], and high stress levels provoked by caloric restriction dieting (CRD) [9], which is the most common diet therapy prescribed for patients who are overweight and obese.

CRD consists of restricting one’s ingested Kcal between 25 and 30% of the daily requirements for maintaining an individual’s ideal body weight [10]. Although patients with obesity following a CRD reduce their body weight, their adherence to treatment is commonly low and present high rates of desertion, most likely due to the elevation in the blood of corticosterone (CORT) levels induced by the caloric restriction; consequently, regaining one’s pre-CRD weight in the following 5 years is very frequent [11]. 

Preclinical studies show that a low food availability is perceived as a stressful challenge; thus, the hypothalamus–pituitary–adrenal (HPA) axis is activated, along with a steady rise in the CORT serum levels [12,13]. This stimulates animals’ appetite by increasing the expression and actions of hypothalamic orexigenic peptides—neuropeptide Y and agouti-related peptide (NPY and AgRP)—as well as reducing the anorexigenic activity of proopiomelanocortin (POMC); as a result, high steroid levels overcome the expected reduction in energy intake of calorie-restricted dieters. Chronic stress with a sustained high CORT is also known to specifically stimulate the intake of high-sugar and fat-containing foods (denominated as highly palatable foods). In fact, a positive emotional change results from the increased intake of palatable foods when the CORT serum levels decrease to their basal values [13]. 

Besides increasing appetite, the chronic elevation of CORT levels triggers the development of anxious behaviors. CORT’s unrestricted entrance to the brain allows the activation of its receptors (glucocorticoid receptor, GR; and mineralocorticoid receptor, MR) in the amygdala, among other areas. Amygdalar stress-responsive neuronal circuitry is deregulated by chronic high brain CORT levels favoring phobic, paranoid, and exacerbated alertness to real or perceived threats, which are components of the symptomatology of anxiety disorders [14,15]. 

The alternative time-restricted feeding regime, in contrast to CRD, does not restrict caloric intake. It consists of an ad libitum access to feeding but only during 8 h of the activity phase of the day, in such a way that the circadian synchronization of feeding is favored; this regime receives the name of intermittent fasting (IF) [16]. As individuals with an IF diet reduce their body weight similarly to those with CRD even when they have no limitation of caloric intake, it raises the question of whether CORT levels increase with a concomitant development of anxiety. 

Among the neuropeptide and neurotransmitter pathways implicated in anxiety development, the amygdala-synthesized thyrotropin-releasing hormone (TRH) is a promising candidate, given that its intraventricular (i.v.) administration reduces rats’ anxiety levels when subjected to the defensive burying behavioral test (DBT) and, also, since TRH mRNA expression in the amygdala is negatively correlated with rats’ anxiety levels analyzed during the same test [17].

Therefore, we aimed to evaluate and compare weekly body weight loss, food intake, and CORT serum levels between rats with 4 weeks of ad libitum or CRD all day (controls, C) and those with ad libitum or CRD offered only during 8 h of their activity phase (IF); also, we evaluated the animals’ anxiety parameters when performing the elevated plus maze (EPM) and their TRH mRNA content in the basolateral nucleus of the amygdala (BLA). 

## 2. Results

### 2.1. Body Weight

The body weight of the four groups was not different at the beginning of the experiment. The control animals showed progressive weight gain along the 4 weeks of this study, starting at 298 ± 10 g on week 1 and finishing at 381 ± 15 on week 4. The IF-adlib and C-CRD groups weighed 11 and 17% less than C-adlib, respectively, only at the week 4, whereas the IF-CRD animals showed 18% and 13% lower weight vs. C-adlib and C-CRD, respectively, on week 2; 21%, 13%, and 13% lower weight than C-adlib, C-CRD, and IF-adlib, respectively, on week 3; and 32%, 15%, and 7% lower weight than C-adlib, C-CRD, and IF-adlib, respectively, on week 4. The two-way repeated measures ANOVA showed a significant effect of diet treatment F_(8,108)_ = 4.169, *p* < 0.01, of time F_(4,108)_ = 101.8, *p* < 0.001, and interaction between both factors F_(12,108)_ = 17.73, *p* < 0.001 (Figure 1A).

### 2.2. Weekly Food Intake

The C-CRD and IF-CRD groups were offered only 30% of the food intake of the C animals (100%). Thus, they showed the exact same trend of feeding: the IF-adlib rats started to reduce their consumption during the first week of the experiment by 20% and also registered 20% less weight on week 3, reaching 26% on week 4 vs. C-adlib (100%). The two-way measures ANOVA showed a significant effect of diet treatment F_(3,392)_ = 14.85, *p* < 0.01, of time F_(4,392)_ = 30.13, *p* < 0.001, and interaction between both factors F_(12,392)_ = 14.85, *p* < 0.001 (Figure 1B).

### 2.3. Corticosterone (CORT) Serum Levels

A Kruskal–Wallis analysis followed by Dunn’s post hoc test showed a 253% and a 120% increase in the CORT serum levels of the C-CRD and IF-CRD animals compared to the controls (100%) respectively. Data are depicted in Figure 2. 

### 2.4. Elevated plus Maze Performance

The effects of the different feeding schedules on the rats’ anxiety levels are depicted in Figure 3. The Kruskal–Wallis analysis followed by Dunn’s post hoc test showed a 48% smaller number of entries of C-CRD rats to the open arms (H_(3)_ = 14.71) but similar entries to the closed ones with respect to C-adlib (100%); the IF-adlib rats entered similarly into the open arms but 48% less in the closed ones (H_(3)_ = 11.50) when compared with C-adlib (100%). The latency to enter the open arms was 150% higher in the CRD group vs. C-adlib (100%) (H_(3)_ = 8.051). We also found a significant and positive correlation between the anxiety levels (number of entries to the open arms) and the CORT serum levels: *p* < 0.05; r^2^ = 0.9173; and Pearson coefficient r = −0.6460.

### 2.5. Pro-TRH mRNA Expression in the Amygdala

TRH is a tripeptide (pyroGlu-his-pro-NH2) whose mRNA sequence is included five times in the high-molecular-weight precursor molecule pro-TRH; thus, we directed the oligonucleotides synthesis and hybridization process to this precursor. In the amygdala slices, the in situ hybridization histochemical study for pro-TRH mRNA expression revealed that, in the basolateral nucleus of this region (BLA), it decreased by 82% in the C-CRD group vs. C-adlib (100%); although that of the IF-adlib and IF-CRD animals was also significantly different to that of C-adlib, the pro-TRH expression was significantly increased vs. that of C-CRD (Figure 4). In Figure 4, the E-H images depict the amplifications of the zone of the amygdala where most of the changes are observed in their correspondent A-D micrographs. We also wanted to include the pattern of pro-TRH expression in the hypothalamic PVN as a positive control of the successful hybridization of the probe, for the signal in the amygdala. The ANOVA showed that F_(3,16)_ = 10.21, *p* < 0.001. 

## 3. Discussion

Circadian rhythms play an important role in maintaining organisms’ homeostasis and, therefore, their physical and emotional states [18]. By sensing the light/dark cycles and through their direct or indirect connections, the hypothalamic suprachiasmatic nucleus (SCN) projections regulate the activity of different brain areas and peripheral tissues in an integrated way and coordinate the rate of catabolic/anabolic cellular pathways with the animals’ activity/resting phases [19]. 

We indeed observed in this study that the IF-adlib rats eating only during 8 h of their activity phase decreased their food intake but showed a similar body weight to the C-adlib group, which supported the notion that food presentation at the same time for some days functioned as a cue which re-trained the circadian regulation of feeding, allowing anticipatory metabolic and behavioral adjustments in that group [20]. Maintaining their body weight even with a reduction in food intake suggested that the IF-adlib animals developed a stronger homeostatic regulation of appetite controlled by fat tissue-producing hormones and a higher energy efficiency than C-adlib. Over a long term, the lower food intake induced by coordinating feeding patterns with the activity phase might help to avoid the energy ingested being stored as fat tissue and body weight gain increases.

Since the food intake of the IF-adlib group was similarly reduced to that of the calorie-restricted animals (C-CRD), it explained why the body weight gain of both groups was not different during this experiment. This agrees with the similar body weight loss observed between patients who are obese or overweight when subjected to CRD or IF regimes [21]; it has also been described that animals [22] or patients following IF [21] present less fat percentage than those with CRD when body composition is analyzed. 

The greater body weight loss of the IF-CRD animals than C-CRD was interesting because both groups were pair-fed over the 4 weeks of this experiment; thus, this supported that, combining caloric restriction with feeding synchronization to the active phase of the day synergized the activation of catabolic pathways’ rate over the anabolic ones, reducing energy storage more efficiently than CRD alone. 

Regarding the stress response, time-restricted feeding was successful in avoiding (in IF-adlib) or attenuating the activation of the HPA axis even when energy intake was restricted (IF-CRD), supporting the notion that the IF animals did not experience the IF regime as a stressful condition, unlike C-CRD. The unstimulated HPA axis and basal or slightly increased CORT serum levels might account for a better adherence of patients who are obese or overweight to an IF regime [23]; this is explained by the effects of CORT on central brain regions, maintaining alertness and increasing anxiety and the discomfort of stress’ negative sensations [24].

A lack of increase in the CORT serum levels in the IF-adlib animals might have prevented their appetite activation by their 16 h of fasting, since they spontaneously decreased their food intake vs. the rats offered food ad libitum all day. High CORT levels increase appetite by enhancing the synthesis and release of orexigenic peptides from the arcuate nucleus [25]. Also, the basal CORT levels might reduce the selection of high-lipid and high-carbohydrate foods in patients following this regime, since we and others have shown that the preference for highly palatable food is increased also by long-term CORT levels’ elevation, which represents a chronic, stressful condition [26]. 

The unresponsiveness of the HPA axis in the IF groups that reduced their food intake “voluntarily” or by being restricted suggested an interference of the afferent information to the hypothalamic PVN coming from the solitary tract nucleus (STN) related to the low nutrient availability in the gut [27]. Likely, the interference might have been specifically towards the NPY-synthesizing neurons from the STN, which it is known to be activated by CRD and, by connecting to the PVN, stimulate corticotrophin-releasing hormone-ergic (CRFergic) cells and elevate the CORT serum levels. In contrast, although the IF-CRD and IF-adlib animals had low amounts of food in their intestines vs. the groups eating all day, the interference of PVN signaling might have been developed by the participation of a central pathway receiving and integrating both metabolic and emotional information, such as the central nucleus of the amygdala (CeA), which is known to stimulate the PVN and the HPA axis directly [15]. It is likely that this nucleus allows or avoids CRFergic neurons’ activation by the fasting-induced stimulation of the STN, depending on the afferent emotional information arriving from the BLA [28]. 

Therefore, we focused on the analysis of a BLA-regulated behavior such as anxiety and on the expression of TRH as an indicator of BLA neuronal activity. TRH is implicated with anxiolytic effects, and, in the amygdala, it is exclusively synthesized in the BLA and potentiates the γ-aminobutyric acid-ergic neurons (GABAergic)’ inhibitory effects of the BLA projections to the CeA [29]. 

We did find a positively correlation between the EPM-evaluated anxiety-like behavior and the CORT serum levels in all the groups, supporting that, in the C-CRD rats, which showed the highest concentration of CORT in the blood along with the highest anxiety levels, HPA axis activation was favored by an enhanced stimulatory CeA input to the PVN, allowing those animals to present a stress response. CeA activity in C-CRD group might have resulted from a basal inhibitory GABAergic neuron input from the BLA. Interestingly, in the same group (C-CRD), we also found that TRH expression in the BLA was reduced when compared with that of the animals that had not been food-restricted (C-adlib); this result supported the notion that this group had a null or a low inhibitory input from the BLA to the CeA and into the PVN.

In contrast, the IF-adlib and IF-CRD that did not show HPA axis stimulation even when food was reduced or restricted did present lower levels of anxiety vs. the groups with all-day food availability, suggesting that their BLA GABAergic inhibitory projections to the CeA were stimulated by the time-restricted intermittent fasting (IF) regime. As indicative of BLA neurons’ activation, we observed a higher TRH mRNA content in that amygdalar nucleus in the IF-adlib and IF-CRD rats, which supported the peptide’s proposed anxiolytic effect [17]; also, it seemed to indicate that the BLA inhibitory connections to the CeA were stimulated, which might have explained the low or blunted response of the HPA axis in those IF groups.

We observed here that the differential changes in HPA axis activation among the groups seemed indirectly depending on the BLA neurons’ stimulation, which, as described above, we identified by the increases in the expression of the anxiolytic-implicated peptide TRH. BLA is known to integrate afferent sensory information from cortical areas and the hippocampus [30], which is critical for animals’ defensive responses to novelty or learned stimuli as they are threatened. Since the animals with IF did not present high CORT levels, it is likely that the synchronization of feeding during the active phase of the day and the resetting of the circadian regulation of appetite by presenting food at the same time over several weeks are cues that induce the expression of clock genes in the BLA, allowing inhibiting GABAergic neurons to be activated, which avoids CeA-PVN connections’ stimulation. 

Although the precise mechanisms for the above were not unraveled, our results pointed out that TRH from the BLA was a putative mediator of this interference. This peptide has been implicated with anxiolytic effects since its i.v. injection reduces the burying time of animals subjected to the DBT test [17]; also, an increment in the amygdala’s TRH mRNA expression has been observed in animals performing the EPM vs. those yoked and has been positively associated with the time spent in the open arms of the maze, which is interpreted as a sign of reduced anxiety-like levels. Also, it is interesting that the role of TRH in mood has been observed both in the instinctive responses to adverse stimuli after a rise in CORT and in learned anxiety processes of recognizing fearful elements in novel environments. In both cases, TRH mRNA expression in the amygdala is elevated by injections of CORT or anxiolytic drugs such as prazosin or ketamine [31,32]. Clinical studies, although scarce, also show that patients with anxiety present lower TRH cerebrospinal levels than the controls [33]. 

## 4. Materials and Methods

All the experiments were conducted in accordance with the Standard Mexican Official NOM-062-ZOO-1999, which regulates the use and care of laboratory animals and was approved by the Ethics Committee of the INPRFM (Number: CEI/C/039/2023). 

Adult male Wistar rats (N = 40) with 250–280 g of body weight obtained from the institutional vivarium were housed in pairs in acrylic cages (34.5 × 49 × 17 cm) and maintained in an inverted light/dark cycle (with light from 19:00 to 7:00 h) and at a controlled temperature (21 ± 1 °C). They were provided with standard laboratory food (Lab rodent diet #5001, PMI Nutrition International, Brentwood, MO, USA) and water ad libitum for a week, ensuring their acclimatization to the laboratory conditions. During this period, we ensured that the rats’ average initial body weight was similar between the groups formed. This study involved four groups of rats (n = 10 animals/group) that were assigned randomly after the acclimation week: a control group (C-adlib), offered food ad libitum all day; C-CRD, offered 30% of C-adlib’s energy requirements all day; IF-adlib, offered food ad libitum between 9:00 and 17.00 h; and IF-CRD, offered 30% of C-adlib’s energy requirements from 9:00 to 17:00 h. The animals were maintained on these feeding schemes for 4 weeks (Figure 5). On the 28^th^ day of the experiment, all the rats performed the EPM, and, after 30 min, they were sacrificed by decapitation. Their brains were extracted, frozen, and stored at −70 °C, until in situ hybridization histochemistry (ISH) analyses were performed; trunk blood was collected for CORT serum levels’ determination.

### 4.1. Body Weight and Food Consumption Registration

During the 4 weeks of this experiment, the average daily food consumption was calculated by each cage with two rats (five cages/group), weighing the remaining amount of food with a digital balance (OHAUS YS series, Mexico City, Mexico) every day at 9:00 h and subtracting it from that offered and weighed the day before at the same time. The grams of food eaten by the C-adlib animals/day were considered as 100%. Thus, the C-CRD and IF-CRD rats were offered only 70% of the Kcal ingested by the C-adlib group. 

We registered the body weight of each animal twice a week using a digital scale. To determine the % of the difference in the body weight/week, we considered the average weight of the rats in C-adlib (n = 10) as 100% each week; thus, the body weight percentage of the difference with that of the C-CRD, IF-adlib, and IF-CRD groups was calculated.

### 4.2. Anxiety-like Behavior Assessment 

The elevated plus maze (EPM) consisted of a cross-shaped labyrinth with four propylene arms radiating from a central platform; the procedure was adapted from a previous work [34]. The apparatus consisted of two “open” arms with 0.5 cm ledges and two “closed” arms with walls of 30 cm height. The open arms were placed opposite each other. The arms were 10 cm wide and 50 cm long and were placed on 55 cm tall wooden legs from the floor. Testing occurred in a quiet environment, under red dim lights (8W Red bulb, Phillips Mexico SA de CV, Mexico City, Mexico). The rats were habituated to the testing room on two separate occasions; the first time was 24 h before the test and the second one 1 h before starting the test. During the test, an animal was placed in the maze’s center facing an open arm and given five minutes to explore the device while being recorded with a camera mounted above the maze (HD recording camera Samsung, HMX-F800; Mexico City, Mexico). The EPM was divided into five zones (two open arms, two closed arms, and a center zone). The parameters used to evaluate the anxiety-like behavior were manually analyzed by two observers. The behaviors typically recorded when rats are in the EPM are the time spent and the entries made into the open arms, closed arms, and center zone. The behavior in this task (i.e., activity in the open arms) reflects a conflict between the rat’s preference for protected areas (e.g., closed arms) and their innate motivation to explore novel environments. The rats’ anti-anxiety behavior (increased open arm time and/or open arm entries, spontaneous motor activity, including total and/or closed arm entries and entry latency (time taken at the start of the test to enter an arm)), were calculated [35,36]. An open arm entry was considered when half of the rat’s body entered to the open arms. The apparatus was cleaned with water and 70% ethanol between the rats.

### 4.3. Corticosterone (CORT) Serum Levels 

For obtaining the serum, the trunk blood of each rat was centrifuged at 3000 rpm for 15 min at 4 °C; 100 μL of serum was used for evaluating the CORT levels as a measure of stress exposure and HPA axis response to different diets and feeding schemes. The analysis was performed by enzyme-linked immunosorbent assay (ELISA), following the manufacturer’s instructions (Enzo Life Sciences Inc., New York, NY, USA), obtaining the absorbance with a microplate reader, and using BioTek’s Gen5 Data Analysis Software (BioTek ELx800, Fisher HealthCare, Houston, TX, USA). The inter-analysis coefficient of variation was 7.8–13.1%, and the intra-analysis coefficient of variation was 6.6–8.0%.

### 4.4. In Situ Hybridization Histochemistry Analysis

Brains being stored at −70 °C were allowed to reach a temperature of −20 °C for 1 h inside a microtome cryostat (Microm HM 525; Carl Zeiss IMT Corp., Maple Grove, MN, USA). Serial coronal sections 14 μm thick were obtained by cryosectioning the brains from −1.084 to −2.28 mm from the bregma, which includes the hypothalamic PVN and the amygdala regions (22), adhered and mounted onto Superfrost/Plus slides (Fisher Scientific, Pittsburgh, PA, USA), and stored at −70 °C. As a positive signal control, we used PVN sections from −1.08 to −1.94 from the bregma. Three slices per slide and three slidesrat were collected. For tissue stabilization and for preventing degradation, the slides were fixed with a 4% paraformaldehyde solution in phosphate buffer (PBS) at 4 °C for 20 min; 24 U/mL of pronase was added to each slide (Calbiochem [Merck Millipore], Darmstadt, Germany) at 20 °C for 2 min and then air-dried before hybridization. A mix of three complementary oligonucleotides was used for hybridization, consisting of the following base sequences of the pro-TRH sequence 105–149, 427–471, 559–603 pair of bases: (i) 50-TGC CCA TGA ATA CTT GTC CTG GTT GGC ACG TCG GCC GGG GTG CTG-30; (ii) 50-ATC AGA CTC CAT CCA GGG GAA GGA TCG CCT GCC AGG GTG CTG CCG-30; and (iii) 50-AGG GTG AAG ATC AAA GCC AGA GCC AGC AGC AAC CAA GGT CCC GGC-30 (Instituto de Biotecnología, UNAM, Cuernavaca, Mexico). These sequences were BLAST tested to evaluate their specificity, which was confirmed using a rat database [NM_013046.2]. The probes were labeled with Digoxigenin-11-dUTP using the activity of the terminal deoxynucleotidyl transferase (Roche Diagnostics GmbH, Mannheim, Germany), as described previously [37]. Two picomoles of labeled oligonucleotides/mL hybridization cocktail of the mixture of three probes were used to hybridize the sections, as described previously [38]; the hybridization cocktail contained 50% formamide, 4× SSC (1× SSC: 150 mM NaCl, 15 mM sodium citrate), Denhardt’s solution (0.02% Ficoll, 0.02% polyvinylpyrrolidone, 0.02% BSA), 10% dextran sulfate, 1% sarkosyl, 20 mM phosphate buffer pH 7.0), 250 µg/mL tRNA of yeast, and 500 µg/mL of salmon sperm DNA. The fixed tissues were incubated with the hybridization solution and covered with parafilm overnight at 41° in an oven. Once the incubation time was over, the sections were washed four times in 0.6 M NaCl and 10 mM Tris-HCl (pH 7.5) (Sigma Chemical Co., St. Louis, MO, USA) at 60 °C for 45 min each time and once in the same buffer at room temperature. After washing, the slides were immersed in a buffer containing 0.1 M Tris-HCl (pH 7.5), 1 M NaCl, 1 mM MgCl2, and 0.5% BSA and incubated at 4 °C overnight with biotin-conjugated anti-digoxigenin-F (ab) fragments (1:200; Roche Diagnostics Mexico, Mexico City, Mexico). The slides were rapidly immersed in the same buffer (without antibody), once in PBS for 10 min, and then mounted with 50% glycerol on 1× PBS. As a negative control, we used a mix of non-complementary probes targeting the same nucleotide sequences of the pro-TRH mRNA. The identification of cells containing TRH mRNA was determined by visualizing the color of the product of the reaction with biotin, and the TRH mRNA signal was considered positive when a brown or blue color was seen on the slice, as previously observed [39].

### 4.5. Image Analysis

The images were observed using a Nikon Eclipse TE 2000-U (Melville, NY, USA); they were captured with a camera and a 10× objective through the NIS-Elements software of Nikon. For each animal, the number of cells within the PVN or the amygdala was quantified and measured in nine rostro-caudal slices containing those regions (Media Cybernetics, Inc., Rockville, MD, USA). 

### 4.6. Statistics

Data are hereby presented as the mean ± SEM. We used two-way repeated measures ANOVA for analyzing body weight and food intake. Food consumption was analyzed considering the cage as an experimental unit (2 rats/cage); when significant, we performed Tukey’s post hoc test. For the other variables, such as entries or latency to open/closed arms, time spent in open/closed arms, and serum CORT concentration, we used Kruskal–Wallis one-way ANOVA; when the *p*-value was significant (*p* < 0.05), we performed Dunn’s post hoc test. For the study of TRH mRNA expression in the amygdala by ISH, a one-way ANOVA was performed, as well as Tukey’s post hoc test when significant. We used the STATA program version 14.0 for our analysis.

## 5. Conclusions

Our results showed that, by synchronizing feeding with the active phase in animals subjected to IF, the calorie restriction-induced response of the HPA axis’ activity and the consequent elevation in the CORT serum levels were attenuated or even blocked. Synchronizing feeding with the activity phase of the day by the IF diet therapy strategy, with food offered ad libitum or even restricted, might decrease or avoid the anguish and anxiety that patients who are obese and overweighed experience when subjected to a CRD. Our study also explains IF’s advantages over the use of CRD to control appetite and favor body weight loss by increasing patients’ adherence to diet therapy.

## Figures and Tables

**Figure 1 ijms-25-05857-f001:**
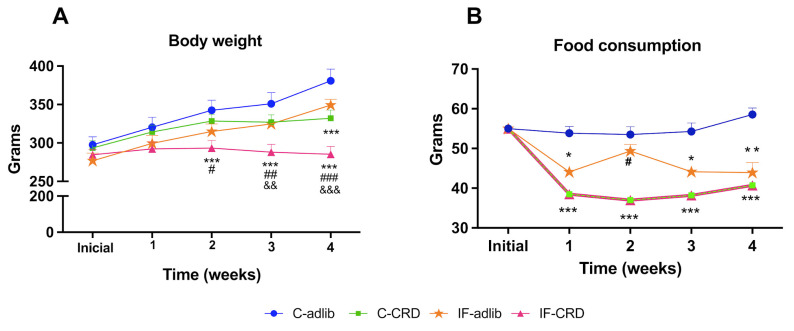
Effect of dietary regime on rats’ body weight and food intake. (**A**) Body weight (body weight) of the control (C) or intermittent fasting (IF) fed ad libitum (adlib) or 30% of the caloric restricted diet (CRD) for 4 weeks (C-adlib, C-CRD, IF-adlib, and IF-CRD), n = 10 rats/group. (**B**) Food consumption of animals is divided into the four groups mentioned above. Data are expressed as the mean ± S.E.M. of the grams of weekly food intake by unit (one cage with two rats; n = 10). * means significantly different vs. C-adlib, ^#^ vs. C-CRD, and ^&^ vs. IF-adlib. Single symbol means: *p* < 0.05; two symbols: *p* < 0.01; and three symbols: *p* < 0.001.

**Figure 2 ijms-25-05857-f002:**
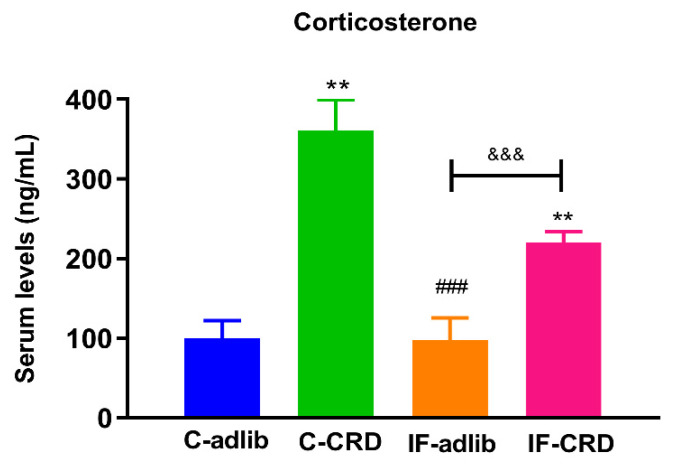
Corticosterone (CORT) serum levels. Data are expressed as the mean ± S.E.M. of the CORT concentration (ng/mL) in the serum. Kruskal–Wallis one-way ANOVA followed by Dunn’s post hoc test showed significant differences between the experimental groups: H_(3)_ = 16.84, *p* < 0.001. Control groups with access to food all day ad libitum (C-adlib) or with a calorie restriction diet (C-CRD); and groups with intermittent fasting (IF, eating only between 9:00 and 17:00 h) offered food ad libitum (IF-adlib) or with a calorie restriction diet (IF-CRD). * means significantly different vs. C-adlib, ^#^ vs. C-CRD, and ^&^ vs. IF-adlib. Two symbols mean: *p* < 0.01; three symbols: *p* < 0.001.

**Figure 3 ijms-25-05857-f003:**
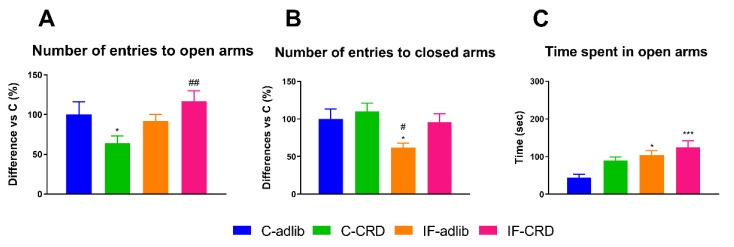
Anxiety-like behavior. Data are expressed as the % of the differences vs. C-adlib of the mean ± S.E.M. of the number of entries into the open arms (**A**) and in the percentage of the difference in the number of entries to the closed arms (C-adlib = 100%) (**B**). The Kruskal–Wallis one-way ANOVA followed by Dunn’s post hoc test showed significant differences between the experimental groups. Panel A: H_(3)_ = 14.71, *p* < 0.01; Panel B: H_(3)_ = 11.50, *p* < 0.01. Control (**C**) groups with access to food all day fed ad libitum (C-adlib) or with a calorie restriction diet (C-CRD); animals eating only between 9:00 and 17:00 h (intermittent fasting, IF), fed ad libitum (IF-adlib), or with a calorie restriction diet (IF-CRD). * when significantly different vs. C-adlib and ^#^ vs. C-CRD. Single symbol: *p* < 0.05, two symbols: *p* < 0.01; and three symbols: *p* < 0.001.

**Figure 4 ijms-25-05857-f004:**
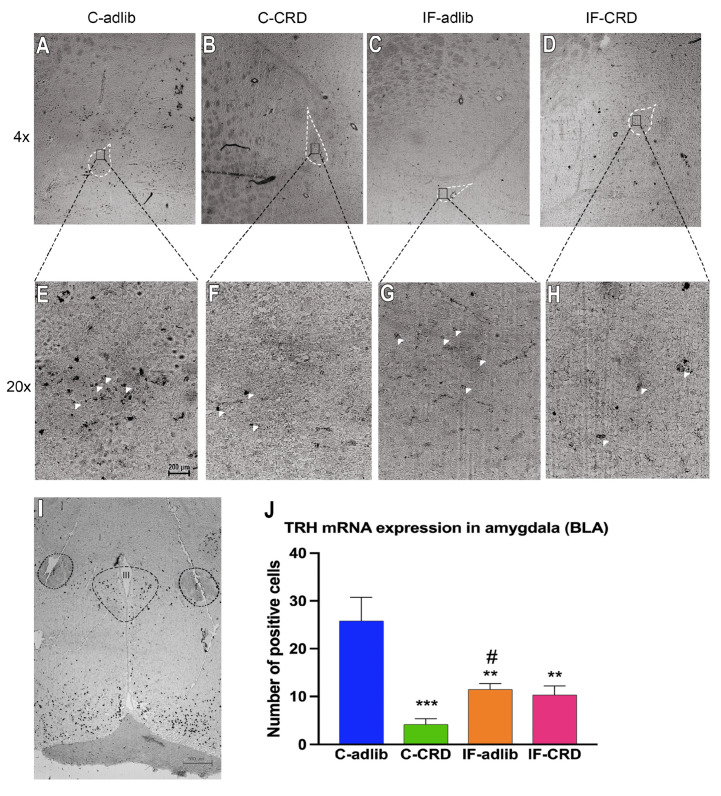
Pro-TRH mRNA expression in the hypothalamic paraventricular nucleus (PVN) and in the amygdala of animals from all groups. mRNAs were detected by in situ hybridization histochemistry analysis using digoxigenin-labeled cDNA probes. (**A**–**D**): Photomicrographs at 4× of coronal slices through the basolateral nucleus of the amygdala (BLA) hybridized with a pro-TRH mRNA probe; (**E**–**H**): amplifications of the observed zone in A to D expressing pro-TRH; (**I**): photomicrograph of a coronal slice through the medial PVN hybridized with a pro-TRH mRNA probe (as a positive control); and (**J**): graphical representation of the densitometric analyses of the TRH mRNA levels with ISH hybridization. Data were analyzed by a one-way ANOVA and a post hoc Tukey’s multiple comparisons test. * means significantly different vs. C-adlib and ^#^ vs. C-CRD, n = 3–5. Single symbol: *p* < 0.05, two symbols: *p* < 0.01; and three symbols: *p* < 0.001. III: third ventricle. Scale bar, 200 μm or 500 µm.

**Figure 5 ijms-25-05857-f005:**
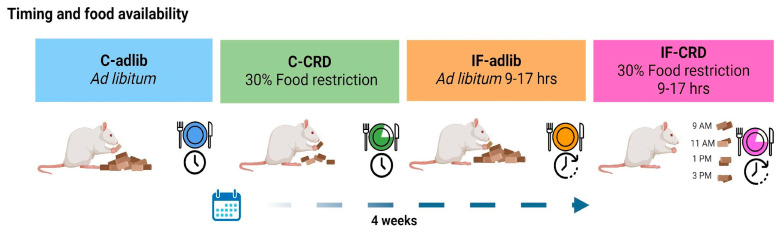
Experimental design for the feeding schedules. Rats were divided into four groups, each one consisting of ten animals. The groups were the following: a control (C-adlib), in which rats had food ad libitum available all day; C-CRD, consisting of rats with 30% food restriction of C-adlib amount of ingested food; IF-adlib, where animals followed intermittent fasting, so they were offered food ad libitum, but with access only from 9 to 17 h; and IF-CRD, rats with intermittent fasting plus 30% food restriction and also time-restricted during their activity phase of the day. The animals were subjected to these feeding schedules for four weeks. Created with BioRender.com.

## Data Availability

Data are available for request.

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
