# Peer review of "Circadian Synchronization of Feeding Attenuates Rats’ Food Restriction-Induced Anxiety and Amygdalar Thyrotropin-Releasing Hormone Downregulation"

_ijms, 2024, doi:10.3390/ijms25115857_

Round 1

Reviewer 1 Report

Comments and Suggestions for Authors

General comments:

The manuscript titled "Circadian synchronization of feeding attenuates rat’s food restriction-induced anxiety and amygdalar TRH downregulation" investigates the impact of circadian synchronization of feeding on anxiety induced by food restriction and the downregulation of amygdalar TRH (thyrotropin-releasing hormone) in rats. The study findings suggest that using intermittent fasting (IF) to restrict feeding time to 8 hours during the active phase can alleviate anxiety symptoms caused by long-term caloric restriction diets (CRD), without the need for caloric restriction, thereby helping to prevent stress and anxiety development. The paper compares corticosterone blood concentrations among rats in control diet, CRD, and IF groups, and evaluates anxiety parameters through the elevated plus maze experiment, while also examining the relationship between amygdalar TRH gene expression and anxiety levels. The results indicate that IF can reverse the anxiety behavior induced by CRD, and the correlation between anxiety levels and TRH expression supports TRH's role in alleviating anxiety. Overall, this paper demonstrates comprehensive and systematic research in study design, experimental procedures, data presentation, and conclusion inference, contributing to its scientific research value. I have some questions and suggestions to improve the quality of the paper.

1.     Some of the abbreviations used in the paper that are not explained, such as PMI. The abbreviations should be accompanied by their full names when first.

2.     The order of the figures needs to be adjusted.

3.     The specifications of the mouse cages should be provided.

4.     The term “cage” and “box" should be consistently.

5.     Meaning of using two-way ANOVA?

6.     Line 274 “4 cages/group” or 5 cages/group.

7.     The statistical presentation and visualization of results could be simplified for better clarity.

8.     Throughout the manuscript, there is inconsistent use of italicization for Latin terms. In some instances, Latin terms that should be in italics are not, and vice versa.

9.       In the manuscript, the referencing style and manuscript format are inconsistent not consistently applied.

10.   Providing page numbers for the references cited.

11.   The purpose of Figures 5A-D and I? They appear to lack of explanations.

Author Response

Reviewer 1

  1. Some of the abbreviations used in the paper that are not explained, such as PMI. The abbreviation should be accompanied by their full names when first.

R: The reviewer is correct; we apologize. We have now included the full name of each term when we first write its abbreviation, along the document.

  1. The order of the figures needs to be adjusted.

R: We have now numbered the figures in the order as they appear in the text.

  1. The specifications of animal cages should be provided

R: The dimensions of the cages are 34.5 x 49 x 17 cm. Now we have included this description in Methods.

  1. The term “cage” and “box” should be used consistently

R: Now we are only using the term “cage”.

  1. Meaning of using two-way ANOVA?

R: As we performed a repeated measure-ANOVA, the time is considered as another factor other than the treatment: Thus, the analysis throws an F value for time, other for diet treatment and their interaction, which we described for body weight and food intake. We are used to name it as a two-way repeated measure-ANOVA.

  1. Line 274 “4 cages/group” or 5 cages/group

R: We now made the correction to 5 cages/group.

  1. The statistical presentation and visualization of results could be simplified for better clarity.

R: We have now tried to simplify the meaning of each symbol and the magnitude of the p value when the differences were significant, in every figure.

  1. Throughout the manuscript, there is inconsistent use of italicization for Latin terms. In some instances, Latin terms that should be in italics are not, and vice versa.

R: We left in italics: in situ, ad libitum, post-hoc

  1. In the manuscript, the referencing style and manuscript format are inconsistent, no consistently applied.

R: Thank you again, we have now revised it and corrected the reference list accordingly to Journal specifications.

.

  1. Providing page numbers for the references cited.

R: Now every reference includes the page numbers.

  1. The purpose of Figures 5A-D and I?. They appear to lack of explanations.

R: We have now included a precise explanation for each image: “In figure 4, the E-H images depicted amplifications of the zone of the amygdala where most of the changes were observed in their correspondent A-D micrographs. We also wanted to include the pattern of pro-TRH expression in the hypothalamic PVN as a positive control of the successful hybridization of the probe, for the signal in the amygdala”. It is highlighted in yellow in the document.

Reviewer 2 Report

Comments and Suggestions for Authors

The paper entitled ‘Circadian synchronization of feeding attenuates rat’s food restriction-induced anxiety and amygdalar TRH downregulation’ aimed to evaluate and compare weekly body weight loss and food intake, corticosterone serum levels between rats with 4 weeks of ad libitum or caloric restriction diet all day vs those with ad libitum or caloric restriction diet only offered during 8 h of their activity phase, along with their anxiety parameters when performing the elevated plus maze also TRH mRNA content in the basolateral nucleus of the amygdala. It was shown that synchronizing feeding with the active phase in animals subjected to intermittent fasting reduced the food restriction-induced response of the HPA axis. This suggests that diet therapy based on an intermittent fasting strategy may decrease the anxiety that obese patients experience when subjected to a caloric-restricted diet. In my opinion, the topic of this work fits well with the scope of IJMS and is quite important and up-to-date because obesity is becoming more and more common in modern society, and nutritional interventions are one of the methods in the treatment of this disease.

The study is generally well designed, however, one wonders why only males were used in these studies, as the results of studies on females could also be interesting. It's also a bit of a pity that the authors didn't decide to assay the expression of NPY, AgRP, and proopiomelanocortin. The manuscript is also well written, however, the text requires author correction due to the same abbreviations being repeatedly introduced in different places in the manuscript. On the other hand, abbreviations used in the figure and the figure’s legend should always be introduced. Additionally, in my opinion, the abstract is missing at least one conclusion sentence.

Author Response

Reviewer 2  

  1. One wonders why only males were used in these studies, as the results of studies on females also be interesting.
  2. We agree with the reviewer because it is a fact that prevalence of anxiety disorders as well as the stress response and deleterious effects of stress are higher in women than in men. Thus it is important to perform these studies in both sexes. We indeed have already performed some experiments with female rats that are subjected to two different regimes of intermittent fasting (fast for 2 days and food offered ad libitum for 5 days/week, 2:5) comparing results in body weight and food intake, and we are obtaining interesting preliminary results.

In the meantime, results obtained here in males regarding low levels of anxiety and HPA axis response when a calorie restricted diet is offered in synchrony with their activity phase of the day will be able to compare to, when those in females are ready.

  1. It’s also a bit of a pity that the authors didn’t decide to assay the expression of NPY, AgRP and proopiomelanocortin.

R: Again, the reviewer is correct, the analysis of appetite regulating neuropeptides would help to explain the changes in food intake found in the different groups of animals. We in fact have already evaluated NPY mRNA levels in the ARC of chronically stressed rats (by isolation) and found that they are lower when are subjected to intermittent fasting than when eating all day (García-Luna et al., 2023).

  1. The manuscript is also well written, however the text requires author correction due to the same abbreviations being repeatedly introduced in different places in the manuscript. On the other hand, abbreviations used in the figure and the figure’s legend should always be introduced.

R: We have now revised the document taking care in writing the correct abbreviations of all terms, in the document and in the figure and figure’s legends.

  1. Additionally, in my opinion, the abstract is missing at least one conclusion sentence.R: We have now included this sentence: “Low anxiety levels induced by synchronizing feeding with light phase would help dieting-patients to improve treatment adherence”